# communications

## biology

# Label-free nanoscale mapping of intracellular organelle chemistry

George E. Greaves [1]✉, Darya Kiryushko[1,2], Holger W. Auner[3], Alexandra E. Porter[2] & Chris C. Phillips [1]✉

The ability to image cell chemistry at the nanoscale is key for understanding cell biology, but many optical microscopies are restricted by the ~(200–250)nm diffraction limit. Electron microscopy and super-resolution fluorescence techniques beat this limit, but rely on staining and specialised labelling to generate image contrast. It is challenging, therefore, to obtain information about the functional chemistry of intracellular components. Here we demonstrate a technique for intracellular label-free chemical mapping with nanoscale (~30 nm) resolution. We use a probe-based optical microscope illuminated with a mid-infrared laser whose wavelengths excite vibrational modes of functional groups occurring within biological molecules. As a demonstration, we chemically map intracellular structures in human multiple myeloma cells and compare the morphologies with electron micrographs of the same cell line. We also demonstrate label-free mapping at wavelengths chosen to target the chemical signatures of proteins and nucleic acids, in a way that can be used to identify biochemical markers in the study of disease and pharmacology.

[1] Experimental Solid State Group, Department of Physics, Imperial College London, London, UK. [2] Department of Materials and London Centre for Nanotechnology, Imperial College London, London, UK. [3] Department of Immunology and Inflammation, The Hugh and Josseline Langmuir Centre for Myeloma Research, Imperial College London, London, UK. ✉email: george.greaves15@imperial.ac.uk; chris.phillips@imperial.ac.uk

The standard route to ultrastructural knowledge, electron microscopy (EM), uses samples that are stained with heavy metals to generate electron absorption contrast and electrical conductivity. They are generally embedded in resin to withstand electron bombardment in a vacuum in a process that takes several days. Furthermore, in the absence of very specific immunolabelling, EM images provide only morphological information, and they only reveal ultrastructure features that take the stain.

A variety of super-resolution fluorescence microscopies also enable sub-diffraction imaging of cellular structures[1,2], but all require labels that are specific to the problem at hand and often rely on a priori knowledge of structures for effective labelling. In practice, photobleaching of the fluorophores, as well as issues with stability and specificity, limits the accuracy to which labels can be localised. Even if such technical challenges are overcome, the ability to localise structures is limited by the distance between the target structure and the fluorophore, set by the lengths of fluorescent labels and linking molecules used, which are typically several nanometres each[3]. Crucially, there is also a risk that labelling perturbs the biology of samples[1].

Infra-red (IR) spectroscopy and imaging are widely used for obtaining quantitative label-free chemical information, but the diffraction limit generally renders it incapable of imaging at the intracellular level. To surpass this limit, several probe-based imaging techniques have been developed which combine the nanoscale resolving power of atomic force microscopy (AFM) with the chemical sensitivity of IR spectroscopy[4,5]. We use scattering-type scanning near-field optical microscopy (s-SNOM), where the AFM probe is illuminated by a tuneable IR laser. Collecting and analysing the IR light backscattered from the small region where the tip and sample interact allows us to infer the IR absorption properties of the sample material in that region. Tuning the laser allows us to image with chemical specificity without the need for staining or labelling. Importantly, we image the biological material itself. We detect ultrastructural features that have never before been imaged directly with light, raising the possibility of finding new intracellular structures that do not show up in EM.

s-SNOM has previously been employed for imaging isolated biological specimens such as protein complexes[6–8], viruses[9], and amyloid fibrils[10], as well as surface imaging of whole cells[11–13]. Intracellular structures in cells and tissue have also been imaged in single-celled organisms[14] and zebrafish retinal tissue[15], respectively. Here we build on this by chemically mapping intracellular structures in human multiple myeloma cells, including, to the authors' knowledge, the first directly optical, sub-diffraction images of endoplasmic reticulum (ER), mitochondria (Mt) and fibrillar components in nucleoli. By varying the illumination wavelength to target different chemical groups in the cells, we also demonstrate the isolation of ultrastructure in a way that's traditionally only achieved with highly specific labelling.

## Results

**s-SNOM setup**. In our s-SNOM setup (Fig. 1), a mid-IR quantum cascade laser (QCL) illuminates a sharp conductive probe which, due to the so-called lightning rod effect, establishes an enhanced and very localised optical field at its tip[16,17]. The presence of the sample modifies this field such that the local IR absorption properties of the sample can be recovered from the backscattered light[18]. In particular, for weak oscillators such as the vibrational modes probed here, the phase of the backscattered light is proportional to the attenuation term of the complex refractive index and is thus also proportional to the far-field absorption coefficient

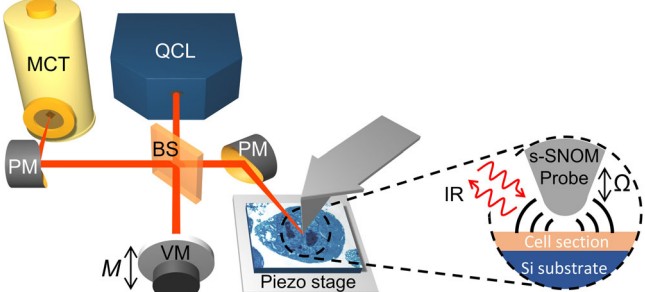

**Fig. 1 s-SNOM set-up.** Infra-red (IR) quantum cascade laser (QCL) light is split into two arms with a beamsplitter (BS). One arm is focused down with a parabolic mirror (PM) to a sharp metallic probe oscillating at frequency $\Omega$ above a sample. The other arm is phase modulated at frequency $M$ by a vibrating mirror (VM) and serves as a reference beam. The backscattered light from the probe and the reference beam is detected with a mercury cadmium telluride (MCT) detector whilst the sample is raster scanned on a piezoelectric stage. The optical properties of the sample are recovered from the backscattered light.

at the particular imaging wavelength[19]. Crucially, the spatial resolution of images is determined by the size of the probing tip, not the wavelength of the light[16,20], allowing us to beat diffraction by typically two orders of magnitude for our imaging wavelength of ~6 μm.

Here we choose to apply the technology to multiple myeloma cells because they are an example of an important human disease where the crucial pathology mechanisms occur at the level of individual cells[21], making them particularly suited to our approach. They are also well-studied with transmission EM (TEM)[22,23]. Cells are fixed, embedded in epoxy resin and sectioned to (70–200) nm but left non-osmicated to enable label-free imaging. Sections used for s-SNOM imaging are placed on a silicon substrate whose reflectivity boosts the signal and improves image quality[24]. Details on the choice of section thickness and the effect of a reflective substrate are provided in Supplementary Notes 1 and 2, respectively. Additional cell sections are poststained with uranyl acetate and lead citrate and imaged with TEM in order to benchmark the s-SNOM images.

Critical to the technique is a pseudo heterodyne detection scheme[25]. The signal measured at the detector is analysed at harmonics of the probe oscillation frequency, $\Omega$, which are split into sidebands separated by the phase modulation frequency, $M$, of the vibrating reference mirror. For the images presented in this paper, the third harmonic of the probe oscillation frequency is used. This yields a highly surface-sensitive, background-free phase measurement, $\phi_3$. The third harmonic is chosen since lower harmonics are generally contaminated by a background signal, whilst higher harmonics exhibit a lower signal-to-noise ratio. A set of images collected at each available harmonic, $n$, is provided in Supplementary Fig. 1 for completeness. In addition to phase measurements, the s-SNOM system also retrieves optical amplitude and sample topography measurements. Representative maps of these quantities and an explanation of the information they contain are provided in Supplementary Fig. 2.

**Nanoscale intracellular imaging of myeloma cells**. Figure 2a is an s-SNOM image of a myeloma cell acquired at 1667 cm$^{-1}$, a wavelength that excites vibrational modes in the amide moieties that are present in proteins and nucleobases. The measured phase shift, $\phi_3$, is proportional to sample absorption and, thus, at this wavelength, amide density. The morphological features in the resulting chemical map enable the identification of several intracellular structures, including a multi-lobed nucleus outlined

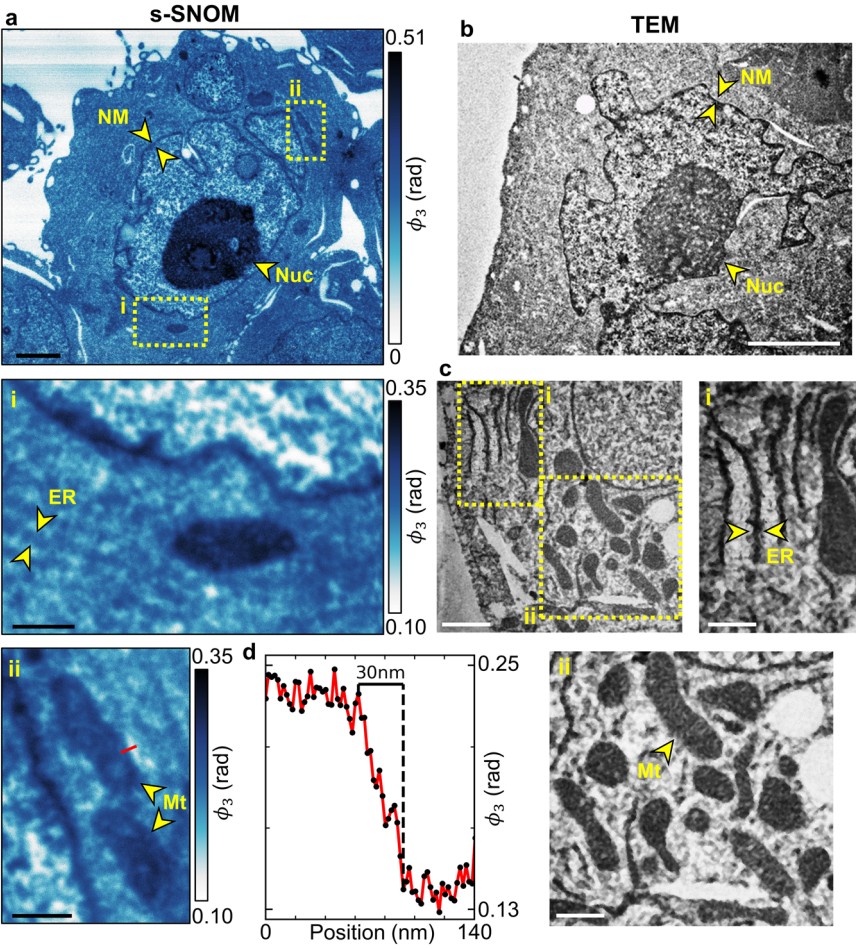

**Fig. 2 s-SNOM chemical mapping and TEM of myeloma cells. a** s-SNOM chemical mapping of myeloma cells acquired at 1667 cm$^{-1}$, targeting amide groups in proteins and nucleobases. Insets **a**(i) and **a**(ii) are higher-resolution s-SNOM images. **b**, **c** TEM of myeloma cells poststained with uranyl acetate and lead citrate but left non-osmicated. Insets **c**(i) and **c**(ii) show structures in greater detail. **d** Spatial profile of the s-SNOM signal along the red line in **a**(ii). NM nuclear membrane, Nuc nucleolus, ER endoplasmic reticulum, Mt mitochondria. Scale bars 2 μm (**a**, **b**), 1 μm (**c**) and 500 nm (**a**(i), **a**(ii), **c**(i), **c**(ii)).

by a nuclear membrane (NM) and containing a nucleolus (Nuc). Similar ultrastructural features are observed in the TEM image of a myeloma cell (Fig. 2b), supporting the identification of structures.

The highest level of amide absorption occurs in the nucleolus. This is the site of ribosome biogenesis, which relies on many proteins, including fibrillarin, nucleolin and nucleophosmin[26–28]. The nuclear membrane is also high in protein[29] and therefore exhibits high amide absorption in the chemical mapping. The granular pattern inside the nuclear envelope is chromatin, made up of DNA tightly wound around histone proteins[30].

Higher resolution s-SNOM images (Fig. 2a (i and ii)) reveal structures that we identify as ER and Mt, visible at this wavelength due to their protein and nucleic acid content[31,32]. A similar morphology is observed in TEM images of ER and Mt in Fig. 2c (i and ii), once again supporting our identification.

Figure 2d shows a line scan of the s-SNOM signal along the red line in Fig. 2aii, which crosses the edge of a mitochondrion. The spatial profile is averaged over a width of 6 nm (3 pixels) and demonstrates a spatial resolution of ≈30 nm. This level of spatial resolution is routinely achieved with these biological samples (Supplementary Fig. 3), but we expect that further optimisation of the imaging technique would see this resolution improved. For example, s-SNOM resolutions down to ~5 nm have been reported with inorganic specimens using sharper bespoke probes[33].

**Multi-wavelength chemical mapping of myeloma cells**. Figure 3a is a far-field absorption spectrum of myeloma cell sections that have been prepared with a formalin-fixed paraffin-embedded (FFPE) protocol, sectioned to a thickness of 1 μm, and subsequently deparaffinised such that the measured absorption does not have a contribution from the embedding medium. We use this far-field absorption spectrum to inform our choice of wavelength for chemical mapping with s-SNOM.

The amide I and II absorption bands are aggregates of many vibrational modes exhibited by amide moieties and are routinely used in IR spectroscopy for analysing the protein content of samples[34], including in nanoscale studies with s-SNOM[10,14]. The phosphodiester (PO$_2$) band corresponds to the PO$_2$ moieties found in the backbones of nucleic acids and phospholipids. Its relative concentration can be used as a reliable biomarker for cancer[35]. The band labelled C–O is predominantly made up of C–O vibrational modes in ribose, the five-carbon sugar in nucleic acids, as well as lipids and carbohydrates. Whilst a variety of macromolecules may also contribute to absorption in the PO$_2$ and C–O bands in different cell regions, they are both routinely used[34] for analysing the nucleic acid content in the nuclear region. We note that much of the lipid content is removed from the cells when they are cleared with xylene, which explains the lack of an absorption peak at approximately 1730 cm$^{-1}$. For completeness, an absorption spectrum for the resin-embedded

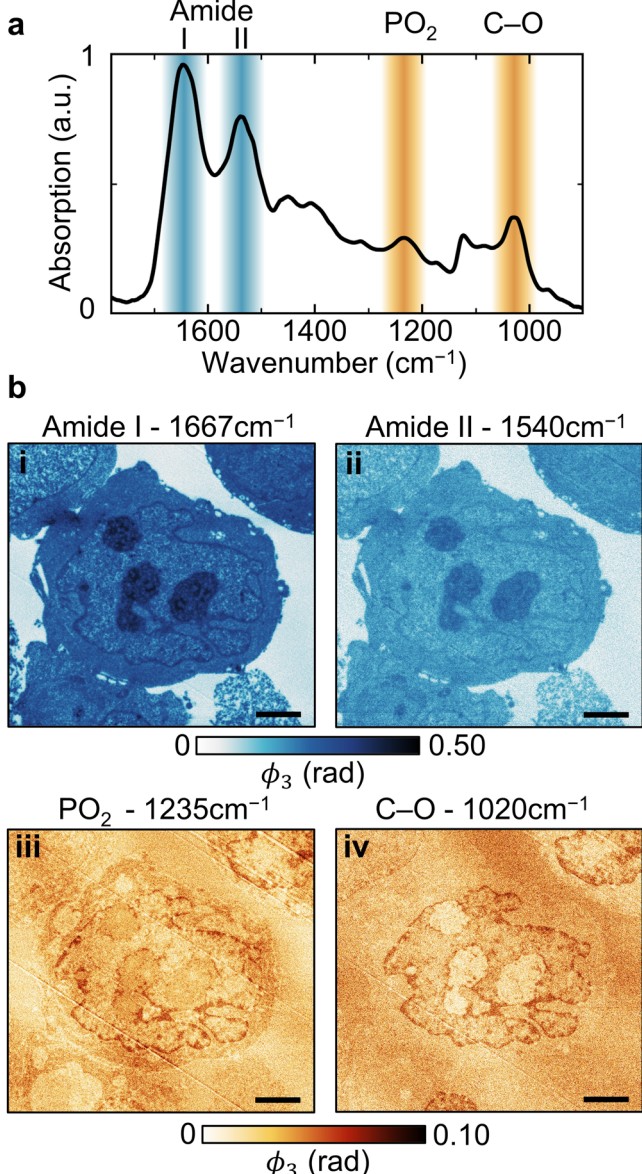

**Fig. 3 Multi-wavelength chemical mapping with s-SNOM. a** Absorption spectrum of deparaffinised formalin-fixed paraffin-embedded (FFPE) myeloma cell sections. **b**(i–iv) s-SNOM images of a single myeloma cell acquired at wavelengths from each of the bands indicated in (**a**). Scale bars 2 μm.

myeloma cells used for s-SNOM imaging is shown in Supplementary Fig. 4.

Figure 3b shows the chemical mapping of a single myeloma cell at wavelengths representing each of the four absorption peaks in Fig. 3a. Figure 3bi, at the amide I peak, serves primarily as a means of mapping the protein density throughout the cell, although there are also contributions from nucleobases. The ubiquity of proteins in the cells results in the entire cell exhibiting strong absorption compared to the embedding resin, which absorbs only weakly at this wavelength.

The strongest absorption is seen in protein-dense structures such as the nucleolus and nuclear membrane, as in Fig. 2a. Figure 3bii also maps amide-containing structures, here using a wavelength in the amide II band. The same structures are seen in the cell but at lower overall absorption levels, corresponding to

the difference in peak heights in the absorption spectrum in Fig. 3a.

Figure 3b (iii and iv) are chemical maps acquired with wavelengths in the $PO_2$ and C–O bands, respectively. At both wavelengths, strong absorption is exhibited close to the nuclear membrane and at the peripheries of nucleoli, which we attribute primarily to densely packaged nucleic acids in chromatin[30]. The cytoplasm and nucleoli exhibit greater absorption at 1235 cm$^{-1}$ than at 1020 cm$^{-1}$. We attribute this to the presence of phospholipids[29], which have many vibronic modes active at 1235 cm$^{-1}$, but far fewer at 1020 cm$^{-1}$ at the edge of the C–O band[34]. We, therefore, surmise that 1020 cm$^{-1}$ is the more suitable imaging wavelength for isolating nucleic acids.

Higher-resolution chemical maps with imaging wavelengths in the amide I and C–O bands are shown in Fig. 4a, b, respectively. This cell appears to be binucleated, as is frequently observed in multiple myeloma cells[36–39], with two protein-dense nucleoli and nuclear membranes visible in Fig. 4a. A higher-magnification image of one of the nucleoli (Fig. 4a (i)) shows several horseshoe-shaped structures with high amide density that surround areas of lower amide density. We identify these as dense fibrillar components (DFCs) and fibrillar centres (FCs), respectively[40]. The rest of the nucleolus is made up of a granular component[40]. High-magnification TEM images of nucleoli (Supplementary Fig. 5) also show DFCs and FCs, supporting our identification. Additional high-magnification images of nucleoli acquired with s-SNOM are shown in Supplementary Figs. 6 and 7. Interestingly, DFCs cannot be seen in Fig. 4b (i), suggesting that they exhibit little absorption at a wavelength of 1020 cm$^{-1}$. This is confirmed in Fig. 4c by the spatial profiles of the s-SNOM signal at both imaging wavelengths across one DFC (magenta line in Fig. 4a (i)).

Remarkably, there are also regions of high nucleic acid density, indicated by cyan arrows in Fig. 4b(i), which are not detected in the amide mapping in Fig. 4a(i), suggesting that they have a low amide density. This is confirmed in Fig. 4d by the spatial profiles of the s-SNOM signal along the green line in Fig. 4b(i). More examples of isolating subsets of cell structure in this way are shown in Supplementary Figs. 6 and 7. In Supplementary Fig. 8, we show that the IR signature of the embedding resin does not noticeably impair our ability to image with chemical specificity.

## Discussion

We have used s-SNOM to perform label-free chemical mapping of eukaryotic cells, combining nanoscale probe-based imaging with the chemical sensitivity of IR spectroscopy. By inspecting the morphology of chemical maps of cells, we have identified many intracellular structures and validated this by comparing them with supporting TEM images. To the authors' knowledge, this paper provides the first direct optical, sub-diffraction images of ER, Mt, DFCs and fibrillar centres in nucleoli.

Further to this, we have mapped proteins and nucleic acids in cells by targeting their well-understood and widely-used IR signatures. This enabled subsets of cell ultrastructure to be isolated, for example, protein-rich DFCs and regions of high nucleic acid content in nucleoli.

Quantification of intracellular structures—in number, size, and chemical composition—is an important aspect of microscopy. Whilst this is possible with labelled cells in super-resolution fluorescence techniques, it has been shown that labelling density affects both the number of structures observed and their apparent size[41]. The label-free nature of s-SNOM, on the other hand, means that chemical mapping can be fully quantitative and comes without such labelling biases. Further, conversion from s-SNOM signals to commonly used optical parameters such as the

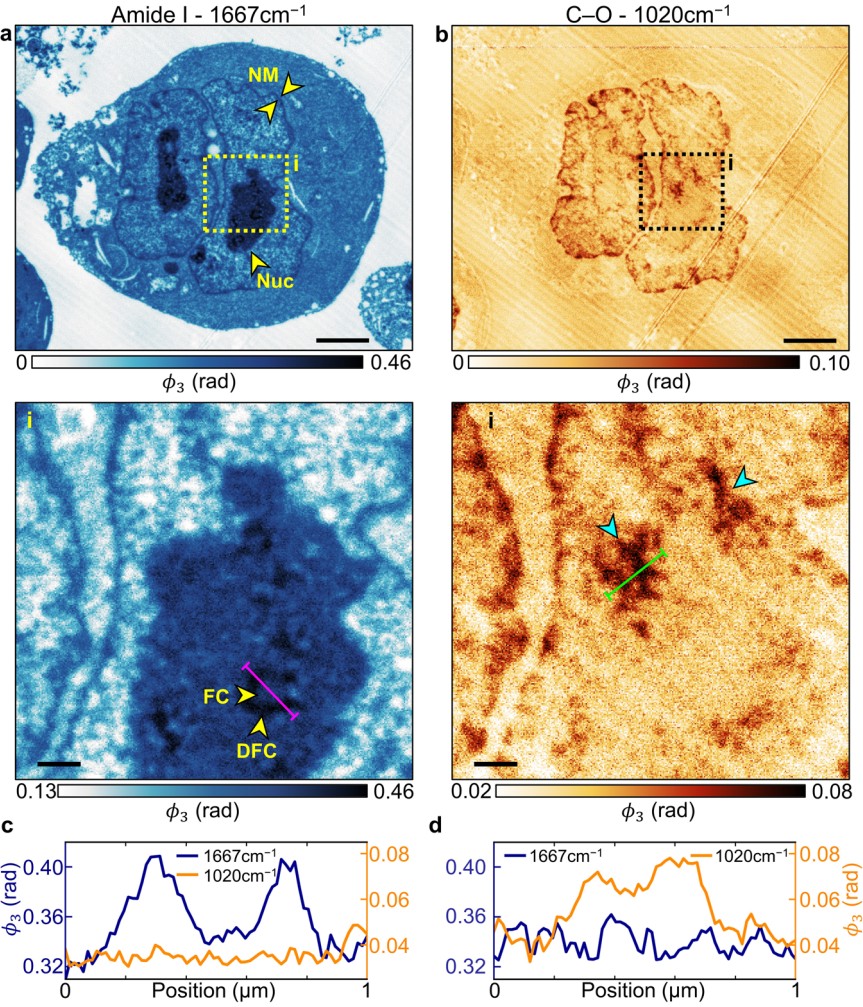

**Fig. 4 High-resolution multi-wavelength chemical mapping.** s-SNOM chemical mapping of a single myeloma cell with imaging wavelengths in the amide I (**a**) and C–O (**b**) bands. Insets **a**(i) and **b**(i) show structures in greater detail. FC fibrillar centre, DFC dense fibrillar component. Spatial profiles of the s-SNOM signal along the magenta (**c**) and green (**d**) lines at both imaging wavelengths averaged over a width of 75 nm (5 pixels). Scale bars 3 μm (**a**, **b**) and 500 nm (**a**(i), **b**(i)).

refractive index is also possible through image processing algorithms[13].

We believe that this capacity for quantitative intracellular imaging offers considerable potential for the discovery of nanoscale biomarkers of disease[42], as well as characterising the efficacy and operating mechanisms of a wide range of therapeutics, in particular drugs such as bortezomib and osimertinib[43], which generally have distinct chemical signatures that can be exploited with s-SNOM. Correlative s-SNOM and light microscopy has already been demonstrated[15], and we expect that correlating additionally with EM is also possible with adaptations of current techniques[44,45]. We, therefore, expect that s-SNOM can complement EM and super-resolution fluorescence techniques to help understand the cellular mechanisms in a variety of research areas, including in immune responses such as NETosis[46].

At our operating wavelength of ~6 μm, our ~30 nm resolution is already ~100 times better than the diffraction limit. Since the maximum achievable spatial resolution in s-SNOM is related to the size of the tip of the scanning probe[20], better resolution can be expected if a sharper tip is used, albeit to the detriment of the image signal-to-noise ratio. Developments leading to improved sensitivity in s-SNOM imaging, for example employing IR nanoantennas[8] or tip-enhancement techniques[47], might help to

alleviate this signal-to-noise issue. We also expect that experimenting with sample preparation schemes that better preserve both the morphological and chemical information would see the image resolution improve towards the ~1 nm reported in inorganic samples[48].

Since s-SNOM is performed in ambient conditions, with only a few milliwatts of IR power, the samples need not be resin embedded, and the technology should be suitable for imaging FFPE style sections[42]. The latter are routinely prepared for histopathology at a fraction of the time and cost of EM samples. This could lead to important clinical applications where giving histopathologists routine access to ultrastructural information without changing the workflow could dramatically improve the monitoring of disease markers.

## Methods

**Sample preparation**. RPMI-8226 cells were grown in an RPMI 1640 medium, supplemented with 10% foetal bovine serum, penicillin (100 U/mL), and streptomycin (100 μg/mL) as performed by Zlei et al.[49]. The cells were purchased from the American Type Culture Collection, verified annually by short tandem repeat analysis and screened for mycoplasma every 3–4 months.

To perform s-SNOM and TEM imaging of RPMI cells, $25 \times 10^6$ cells were pelleted, washed in 0.1 M 4-(2-hydroxyethyl)−1-piperazineethanesulfonic acid (HEPES) buffer (pH 7.2), fixed with 2.5% glutaraldehyde in HEPES for 2 h at 4 °C, and washed 3 × 10 min with HEPES. Thereafter, pellets were dehydrated in a

graded ethanol concentration (50%, 70%, 95%, and dry 100% v/v), 3×5 min, followed by 3×10 min in acetonitrile (Sigma-Aldrich, UK). The samples were then progressively infiltrated with 50%, 75%, and 100% solutions of Quetol resin (Sigma) in acetonitrile at RT (2 h × 50% resin, overnight × 75% resin, and 2× changes of 100% resin over three days) and cured at 60 °C for 24 h. Selected samples were post-stained with 5% uranyl acetate (UA, 5 min) and lead citrate (LC, 3%, 5 min), both made up in double-distilled water. The embedded pellets were cut using a Leica UC7 ultramicrotome (Leica, Austria) with a 35° diamond knife (Diatome, Switzerland) to a thickness of (70–200) nm. Sections were immediately collected on carbon-coated TEM copper grids (Agar Scientific, UK) or on silicon wafer chips (NanoAndMore), then dried and stored until further use.

To obtain a far-field absorption spectrum, RPMI-8226 cells were prepared with a standard FFPE protocol. A Leica 1400 microtome (Leica, Germany) with a 22° stainless steel blade (S22, Feather, Japan) was used to cut sections to a thickness of 1 μm. Sections were placed on a calcium fluoride slide and deparaffinised with xylene (2 × 3 min) followed by ethanol (100%, 2 × 3 min).

**s-SNOM**. A commercial near-field microscope system (neaSNOM, NeaSpec, Germany) equipped with a QCL system (MIRcat-QT, Daylight Solutions, USA) with four lasing chips covering a spectral range of ∼(900–1900) cm$^{-1}$ was used for s-SNOM imaging. A pseudoheterodyne detection scheme[25] was employed to obtain background-free phase measurements. Commercially available probes (Arrow NCPt, NanoWorld, Switzerland) with resonant frequency ∼285 kHz were driven in tapping mode with ∼50 nm amplitude. Gwyddion was used for basic image processing, such as line noise correction and extracting line profiles from image data.

**Transmission EM**. TEM imaging of myeloma cells was carried out at 120 kV (JEOL 2100 TEM, JEOL).

**Far-field absorption spectrum**. The far-field absorption spectrum of myeloma cells was obtained with a commercial Fourier transform IR spectrometer (Vertex 70, Bruker, USA) with Hyperion IR microscope attachment, operated in transmission mode.

**Reporting summary**. Further information on research design is available in the Nature Portfolio Reporting Summary linked to this article.

## Data availability
s-SNOM and TEM images are available at the BioImage Archive repository with accession code S-BIAD612. Raw absorption spectra data are available at https://doi.org/10.6084/m9.figshare.22277086.v1. Supplementary information is provided in a separate document.

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

## Acknowledgements

C.C.P. and G.E.G. acknowledge financial support from EPSRC (EP/N509486/1, EP/R513052/1). H.W.A. acknowledges the support of Cancer Research UK (C41494/A29035). C.C.P. and D.K. acknowledge the support of Cancer Research UK (C68186/A28503). The authors thank Sandra Iles for producing the FFPE block used for obtaining the far-field absorption spectrum of multiple myeloma cells.

## Author contributions

The project was conceived and coordinated by C.C.P. and A.E.P. Myeloma cells donated by H.W.A. were prepared into sections by D.K. s-SNOM images were acquired and analysed by G.E.G. TEM images were acquired by D.K. Myeloma FFPE blocks were prepared for use in an FTIR spectrometer and subsequently measured by G.E.G. The paper was written by G.E.G. with contributions from other authors. All authors have read and acknowledged the paper.

## Competing interests

The authors declare no competing interests.
