## [Peer Review File · Communications Biology]

Reviewers' comments:

Reviewer #1 (Remarks to the Author):

In the manuscript, the authors reported an intracellular nanoscale chemical study using s-SNOM. Majorly the distribution of protein and nucleic acid was revealed within organelles. Overall, the manuscript is well organized, and the data is generally clearly presented. However, I did find several issues that need to be addressed before publication.

Major issues:

1. s-SNOM is capable of performing both chemical mapping and point spectroscopy. Surprisingly, the authors did not include any near-field IR spectrum in the manuscript. A far-field FTIR spectrum cannot represent chemical information when it goes down to the nanoscale.

Lack of s-SNOM spectra yields ambiguity in data interpretation. For example, in Figure 3, the authors attributed the greater absorption at 1235 cm^{-1} to phospholipids in the cytoplasm and nucleoli. This assumption can be easily verified by showing a point spectrum at cytoplasm and nucleoli. It is new to see phospholipids in nucleoli. Another example is Figure 4. A signal cross section only represents a contrast within the area and cannot be interpreted as whether a chemical is present or not. The authors' statement in Line 118 that DFCs contain a relatively low density of nucleic acids is not convincing. An IR spectrum is indeed necessary to show the actual chemical composition.

2. In figure 3, the FTIR spectrum does not show a lipid peak at around 1730 cm^{-1} . It would be helpful to explain the reason for this missing peak.

3. The cell slices were embedded in epoxy resin, and the authors need to demonstrate that resin is not altering/contributing to the IR assignments of major components, i.e., protein and nucleic acids. The IR spectrum of resin should be included as a reference.

4. From my personal point of view, the biological findings in this manuscript seems well-known knowledge to the community. The authors may want to add more innovative discoveries to make the whole story stronger. For example, the authors mentioned that nanoscale chemical composition can serve as biomarkers of diseases. Then it would be interesting to compare normal cells and cancer cells in terms of protein/lipid ratios or other biomarkers.

Minor issues:

1. In Line 34, reference No. 4 is a TERS study, not a combination of AFM and IR. The authors need to either remove or replace this reference.

2. The authors claimed in the Abstract (Line 17) that quantitative chemical mapping was done, which is not true. The authors only showed qualitative assignments and comparisons in the manuscript. So this over-claim should be removed.

Reviewer #2 (Remarks to the Author):

The authors report a study on s-SNOM imaging of eukaryotic cells. Using a tunable QCL they use excitation lengths aligned to various biochemical constituents of the cells, showing that label-free s-SNOM imaging can be very specific in imaging particular organelles and sub-cellular structures. While

the results are not surprising and could be anticipated based on previous works addressing prokaryotic cells, such as the work of [Kanevche, K., *Commun Biol* 4, 1341 (2021)] or [Mészáros, Let al., *Chemical Science* 11.18 (2020): 4608-4617.], they represent a premiere with respect to s-SNOM imaging of eukaryotic cells and will surely be of interest, at least to the s-SNOM community. While the authors mainly focus on comparing s-SNOM and EM outputs, showing correspondences between the two, this work would have enjoyed additional weight if more emphasis would have been placed on showcasing cell parts that cannot be resolved with other techniques, if any. Overall though, this work will serve as a useful resource and will surely inspire other efforts in this direction.

Specific comments:

1) Line 10: "many optical microscopies are restricted by the ~250nm diffraction limit". I think the diffraction limit is actually ~200nm, not ~250nm. Take for example CLSM imaging under 405nm illumination.

2) In some parts of the manuscript the language quality lacks consistency. For example, Line 28: "In practice photobleaching of the fluorophores, as well as issues with stability and specificity, limits the accuracy to which they can be localised, and the fluorophores themselves (and any linking molecules used) can be several nanometres in length.". Some punctuation marks are missing, and the end of the sentence seems unrelated to the rest. The authors should carefully re-read the whole text, and improve the manuscript from this point of view.

3) Line 31: "[...] Furthermore, knowledge of structures is required in order to correctly label them, meaning fluorescence microscopies are not discovery techniques". Indeed apriori knowledge on the sample composition can be of great benefit to fluorescence super-resolution microscopy, but I don't think this renders them as techniques that are not suited for "discovery". There are many studies where these techniques have in fact enabled important discoveries, so I think this statment should be re-evaluated.

4) Line 40: "Tuning the laser allows us to perform IR spectroscopy at the nanoscale without the need for staining or labelling. Importantly, we image the tissue itself". With respect to the latter sentence, if nothing escapes my attention, this study focuses on cell imaging, not tissue imaging, so this statment seems ill-suited.

5) Line 42: "s-SNOM has previously been employed for imaging isolated biological specimens, such as protein complexes^{6,8}, viruses⁹, single-celled organisms¹⁰ and amyloid fibrils¹¹". Besides these, s-SNOM has been used also for tissue imaging [Stanciu S.G., *Biomed. Opt. Express* 8, 5374-5383 (2017)]. The samples used in that study have been in fact prepared using an EM protocol based on ultra-thin sections.

6) Line 54: "For the images presented in this paper, the s-SNOM signal is demodulated at the third harmonic of the probe oscillation frequency". Why was this particular harmonic used ? Was it better than the others ? Or was there no significant difference between 2-5th harmonics that the NeaSpec system typically records ? This should be clarified. Furthermore, a Figure with a relevant image collected at all harmonics should be provided at least in the Supplementary.

7) Line 60: "Cells are fixed, embedded in epoxy resin and sectioned to (70–200)nm but left non-osmicated to enable label free imaging". Could the authors explain the large range of section thickness (70-200nm) ? Is it because the employed ultramicrotome is not consistent in its outputs ? Or cells sections of different thicknesses were voluntarily used for different segments of this study ?

8) Line 61: " Sections used for s-SNOM imaging are placed on a silicon substrate whose reflectivity boosts the signal and improves image quality²¹". I think that it is generally acknowledged that the volume that contributes to the recorded s-SNOM signals is in the range of 20-100nm. Does this differ for the samples here in discussion ? For example, does the Si substrate have any effect on attainable s-SNOM contrast/signal quality for 200nm sections ?

9) Line 66: "The measured phase shift, ϕ_3 , is proportional to sample absorption and thus, at this wavelength, amplitude density.". The results are in line with what has been reported in other works on s-SNOM. Tuning the illumination in the absorption band of a sample constituent, results in phase contrast from it. However, given that this article does not target only the s-SNOM community but also the life-science community who was scarcely exposed to s-SNOM concepts so far, it would be important that the authors provide a brief description on the physics principles that explain why in s-SNOM you get phase contrast from a sample constituent illuminated in its absorption band. With respect to this request, the authors may be tempted to simply refer to other works (for example those that experimentally showed that the far-field absorption spectra matches the s-SNOM phase spectra), but this is not what I am asking. It is actually not easy to find in the literature an exact explanation on this aspect, and it would be very useful for the targeted readership if the authors can provide it here.

10) Line 81: "but s-SNOM resolutions down to ~ 5 nm have been reported with inorganic specimens²⁹ and further optimisation of the imaging technique may help us reach that level". Could the authors clarify if they tried to perform imaging with sharper probes such as those used in the works they refer to ? Why did they stop at 30nm resolution ? Is it simply because this was the size of the probes available in their lab when performing this work ? Or maybe there is some reason related to the studied sample ?

11) Line 113: " The cell appears to be binucleated, with two protein-dense nucleoli and nuclear membranes visible in figure 4 (a)". Are myeloma cells usually binucleated ? Or this is something that is seldom observed ?

12) Line 126: "By inspecting the morphology of chemical maps of cells, and comparing with supporting TEM images, we have identified many intracellular structures that have not been directly optically imaged before." With respect to the last part, it would be useful to add a table or paragraph, and summarize what cell parts discussed in this manuscript have been resolved with other optical techniques, and which haven't.

13) Line 140: "We note that s-SNOM imaging should be suitable for samples that have been prepared for correlative light and electron microscopy³⁶, in particular resin-embedding and cryosectioning approaches." s-SNOM imaging has actually been already demonstrated on samples prepared for correlative light and electron microscopy. In [Stanciu S.G., Biomed. Opt. Express 8, 5374-5383 (2017)] the authors discussed correlative s-SNOM -CLSM imaging on fluorescently labeled ultra-thin zebrafish retina sections cut with an ultramicrotome, according to correlative fluorescence-EM protocols [J. M. Mateos, Sci. Rep. 6, 34062 (2016)].

14) Line 146: "Since the maximum achievable spatial resolution in s-SNOM is related to the size of the tip of the scanning probe¹⁷, better resolution can be expected if a sharper tip is used, albeit to the detriment of image signal-to-noise ratio. Developments leading to improved sensitivity in s-SNOM imaging, for example using 10 infrared nanoantennas⁸, might help to alleviate this signal-to-noise issue". Recently a study introduced scanning probe tips with tunable optical properties [Belhassen, J., et al. Advanced Photonics Nexus 2.2 (2023): 026002.]. Would such tips be of benefit with respect to higher SNR in this bioimaging application here discussed ?

15) I think not all figures in the supplementary are referenced in the text. The authors should check.

16) It would be relevant if the authors could comment on how qualitative interpretation of s-SNOM phase images collected on myeloma cells could potentially be complemented by quantitative s-SNOM (refractive index) imaging, which has been previously demonstrated on red blood cells. [Tranca, D.E. *Nanomedicine* 14.1 (2018): 47-50].

17) The authors do not discuss at all amplitude s-SNOM images, nor AFM images. Are s-SNOM amplitude images useless in this application ? At least in the supplementary a full set (s-SNOM amplitude, s-SNOM phase, AFM) should be displayed. Also, it would be interesting to discuss how s-SNOM amplitude images depend on illumination wavelength. Can this dependence be physically modelled ? In the paragraph that the authors will add to the paper according to point 9 earlier, they may also refer to dependence on s-SNOM amplitude images with wavelength.

18) In [Kanevche, K., *Commun Biol* 4, 1341 (2021)] the authors showed 3D s-SNOM imaging of prokaryotic cells, based on imaging of consecutive sections. This would be very nice to add to this work.

19) It would be very useful for the s-SNOM community if the authors upload the whole raw dataset collected in this experiment as a public resource (e.g. on Data Dryad, OSF, FigShare, etc) and add it as a reference.

Reviewer #3 (Remarks to the Author):

In this manuscript, the authors apply the infrared nanoscopy technique of s-SNOM to measure tissues. IR s-SNOM obtained chemical contrast that regular electron microscopy cannot provide. The author demonstrated IR imaging of tissue based on several IR absorption bands of biological molecules. The results have their merits in the biological application of s-SNOM, which is a tool that has just started to gain traction in the realm of biology. The work is publishable.

One minor comment is that the authors used a commercial instrument of NeaSNOM, and stated that third harmonic demodulation was used. However, regular third harmonic demodulation does not give a phase value corresponding to the IR absorption. The authors must have used the technique of pseudoheterodyne, which is available in NeaSNOM. The author should update their method description to correctly state this technical detail. For those who do not have NeaSNOM, this information would be critical for repeatability.

COMMSBIO-23-0150-T Reviewer comments response

Reviewer comments	Response to reviewer
Reviewer #1 (Remarks to the Author):	
In the manuscript, the authors reported an intracellular nanoscale chemical study using s-SNOM. Majorly the distribution of protein and nucleic acid was revealed within organelles. Overall, the manuscript is well organized, and the data is generally clearly presented. However, I did find several issues that need to be addressed before publication.	We thank the referee for their thorough review of our manuscript and many constructive suggestions. Please see below for the actions we have taken to address the comments.
Major issues:	
1. s-SNOM is capable of performing both chemical mapping and point spectroscopy. Surprisingly, the authors did not include any near-field IR spectrum in the manuscript. A far-field FTIR spectrum cannot represent chemical information when it goes down to the nanoscale. Lack of s-SNOM spectra yields ambiguity in data interpretation. For example, in Figure 3, the authors attributed the greater absorption at 1235 cm⁻¹ to phospholipids in the cytoplasm and nucleoli. This assumption can be easily verified by showing a point spectrum at cytoplasm and nucleoli. It is new to see phospholipids in nucleoli. Another example is Figure 4. A signal cross section only represents a contrast within the area and cannot be interpreted as whether a chemical is present or not. The authors' statement in Line 118 that DFCs contain a relatively low density of nucleic acids is not convincing. An IR spectrum is indeed necessary to show the actual chemical composition.	“Surprisingly, the authors did not include any near-field IR spectrum in the manuscript” We agree that this would be a great addition to this work. Unfortunately, this isn't possible with our setup since we do not have the required broadband source and nano-FTIR module. “A far-field FTIR spectrum cannot represent chemical information when it goes down to the nanoscale” We accept this comment. Given that the technical limitations discussed above mean we cannot obtain a nano-FTIR spectrum, we feel strongly that the far-field spectrum should be included as it was used to inform our choice of imaging wavelength. To avoid any doubt in its purpose, we have updated the manuscript (line 103):“We use this far-field absorption spectrum to inform our choice of wavelength for s-SNOM imaging”. “The authors' statement in Line 118 that DFCs contain a relatively low density of nucleic acids is not convincing” We understand and accept that the reviewer would wish to see nano-FTIR spectra to be convinced. Given the technical limitations discussed above, we are happy to instead revise the claim. See updated manuscript (line 140): “Interestingly, DFCs cannot be seen in figure 4 (b) (ii), suggesting that they exhibit little absorption at a wavelength of 1020cm⁻¹.”
2. In figure 3, the FTIR spectrum does not show a lipid peak at around 1730 cm⁻¹. It would be helpful to explain the reason for this missing peak.	We thank the reviewer for the suggestion. We have revised the manuscript to address this point (line 112): “We note that much of the lipid content is removed from the cells when they are cleared with xylene, which explains the lack of an absorption peak at approximately 1730cm⁻¹.”
3. The cell slices were embedded in epoxy resin, and the authors need to demonstrate that resin is not altering/contributing to the IR	We accept this comment and gladly supply the IR spectrum of resin in Supplementary Figure

assignments of major components, i.e., protein and nucleic acids. The IR spectrum of resin should be included as a reference.	4 and comment on its contribution in the caption.
4. From my personal point of view, the biological findings in this manuscript seems well-known knowledge to the community. The authors may want to add more innovative discoveries to make the whole story stronger. For example, the authors mentioned that nanoscale chemical composition can serve as biomarkers of diseases. Then it would be interesting to compare normal cells and cancer cells in terms of protein/lipid ratios or other biomarkers.	This is an excellent suggestion and is indeed how we are likely to proceed. However, a study such as this is likely to take several years and is not in the scope of the current project.
Minor issues:	
1. In Line 34, reference No. 4 is a TERS study, not a combination of AFM and IR. The authors need to either remove or replace this reference.	We accept that the statement was ambiguous and have replaced the reference to avoid any confusion.
2. The authors claimed in the Abstract (Line 17) that quantitative chemical mapping was done, which is not true. The authors only showed qualitative assignments and comparisons in the manuscript. So this over-claim should be removed.	We have revised the manuscript to include an explanation of how the s-SNOM signal relates to absorption, but we respect that this could still be seen as an over-claim and have removed it from the abstract.
Reviewer #2 (Remarks to the Author):	
The authors report a study on s-SNOM imaging of eukaryotic cells. Using a tunable QCL they use excitation lengths aligned to various biochemical constituents of the cells, showing that label-free s-SNOM imaging can be very specific in imaging particular organelles and sub-cellular structures. While the results are not surprising and could be anticipated based on previous works addressing prokaryotic cells, such as the work of [Kanevche, K., Commun Biol 4, 1341 (2021)] or [Mészáros, Let al., Chemical Science 11.18 (2020): 4608-4617.], they represent a premiere with respect to s-SNOM imaging of eukaryotic cells and will surely be of interest, at least to the s-SNOM community. While the authors mainly focus on comparing s-SNOM and EM outputs, showing correspondences between the two, this work would have enjoyed additional weight if more emphasis would have been placed on showcasing cell parts that cannot be resolved with other techniques, if any. Overall though, this work will serve as a useful resource and w'll surely inspire other efforts in this direction.	We thank the reviewer for taking the time to review our manuscript and for the many constructive comments. Please see below for the actions we have taken to address the comments and suggestions.
Specific comments:	

1) Line 10: "many optical microscopies are restricted by the ~250nm diffraction limit". I think the diffraction limit is actually ~200nm, not ~250nm. Take for example CLSM imaging under 405nm illumination.	We thank the reviewer for pointing this out. We have revised the manuscript to account for this example (line 10): "the ~(200–250)nm diffraction limit".
2) In some parts of the manuscript the language quality lacks consistency. For example, Line 28: "In practice photobleaching of the fluorophores, as well as issues with stability and specificity, limits the accuracy to which they can be localised, and the fluorophores themselves (and any linking molecules used) can be several nanometres in length.". Some punctuation marks are missing, and the end of the sentence seems unrelated to the rest. The authors should carefully re-read the whole text, and improve the manuscript from this point of view.	We accept the reviewer's suggestion and have revised the paragraph containing this sentence (lines 26–32), as well as other parts of the manuscript.
3) Line 31: "[.] Furthermore, knowledge of structures is required in order to correctly label them, meaning fluorescence microscopies are not discovery techniques". Indeed apriori knowledge on the sample composition can be of great benefit to fluorescence super-resolution microscopy, but I don't think this renders them as techniques that are not suited for "discovery". There are many studies where these techniques have in fact enabled important discoveries, so I think this statment should be re-evaluated.	We accept this comment and have revised this sentence to soften the claim (line 27): "... and often rely on a priori knowledge of structures for effective labelling".
4) Line 40: "Tuning the laser allows us to perform IR spectroscopy at the nanoscale without the need for staining or labelling. Importantly, we image the tissue itself". With respect to the latter sentence, if nothing escapes my attention, this study focuses on cell imaging, not tissue imaging, so this statment seems ill-suited.	We thank the reviewer for pointing this out and have revised the manuscript (line 40), with 'biological material' in place of 'tissue'.
5) Line 42: "s-SNOM has previously been employed for imaging isolated biological specimens, such as protein complexes^{6,8}, viruses⁹, single-celled organisms¹⁰ and amyloid fibrils¹¹". Besides these, s-SNOM has been used also for tissue imaging [Stanciu S.G., Biomed. Opt. Express 8, 5374-5383 (2017)]. The samples used in that study have been in fact prepared using an EM protocol based on ultra-thin sections.	We accept the suggestion to refer to this work and have revised the manuscript accordingly (line 45).
6) Line 54: "For the images presented in this paper, the s-SNOM signal is demodulated at the third harmonic of the probe oscillation frequency". Why was this particular harmonic used ? Was it better than the others ? Or was there no significant difference between 2-5th	We have updated the manuscript to address this point (line 73): "The third harmonic is chosen since lower harmonics are generally contaminated by a background signal, whilst higher harmonics exhibit a lower signal-to-noise ratio. A set of images collected at each

harmonics that the NeaSpec system typically records ? This should be clarified. Furthermore, a Figure with a relevant image collected at all harmonics should be provided at least in the Supplementary.	available harmonic, n, is provided in Supplementary Figure 1 for completeness". As stated, we have included images acquired at different harmonics in Supplementary Figure 1.
7) Line 60: "Cells are fixed, embedded in epoxy resin and sectioned to (70–200)nm but left non-osmicated to enable label free imaging". Could the authors explain the large range of section thickness (70-200nm) ? Is it because the employed ultramicrotome is not consistent in its outputs ? Or cells sections of different thicknesses were voluntarily used for different segments of this study ?	We thank the reviewer for inviting us to provide clarity here. We have added to the supplementary information an explanation for the range of section thickness (supplementary note 1).
8) Line 61: " Sections used for s-SNOM imaging are placed on a silicon substrate whose reflectivity boosts the signal and improves image quality²¹". I think that is generally acknowledged that the volume that contributes to the recorded s-SNOM signals is in the range of 20-100nm. Does this differ for the samples here in discussion ? For example, does the Si substrate have any effect on attainable s-SNOM contrast/signal quality for 200nm sections ?	We have added to the supplementary information an explanation of why we see an improved signal even when the section thickness is greater than contributing volume that the reviewer correctly refers to here (supplementary note 2)
9) Line 66: "The measured phase shift, ϕ_3, is proportional to sample absorption and thus, at this wavelength, amide density.". The results are in line with what has been reported in other works on s-SNOM. Tuning the illumination in the absorption band of a sample constituent, results in phase contrast from it. However, given that this article does not target only the s-SNOM community but also the life-science community who was scarcely exposed to s-SNOM concepts so far, it would be important that the authors provide a brief description on the physics principles that explain why in s-SNOM you get phase contrast from a sample constituent illuminated in its absorption band. With respect to this request, the authors may be tempted to simply refer to other works (for example those that experimentally showed that the far-field absorption spectra matches the s-SNOM phase spectra), but this is not what I am asking. It is actually not easy to find in the literature an exact explanation on this aspect, and it would be very useful for the targeted readership if the authors can provide it here.	We agree with this suggestion and note that the reviewer is correct to say that an explanation is hard to find as the physical insight is very much buried in the maths of various analytical models and detection schemes. We have included the sentence (line 56) "In particular, for weak oscillators such as the vibrational modes probed here, the phase of the backscattered light is proportional to the attenuation term of the complex refractive index and is thus also proportional to the far-field absorption coefficient at the particular imaging wavelength¹⁹". This summarises the explanation in Huth, F et al. (2012). Nano Lett. 12, 3973-3978, which we believe to be the best explanation of this aspect.
10) Line 81: "but s-SNOM resolutions down to ~5nm have been reported with inorganic specimens²⁹ and further optimisation of the imaging technique may help us reach that	We thank the reviewer for inviting us to be clearer here. The probes used in the reference were homemade. We have revised the manuscript to make this point (line 98): "For

level". Could the authors clarify if they tried to perform imaging with sharper probes such as those used in the works they refer to ? Why did they stop at 30nm resolution ? Is it simply because this was the size of the probes available in their lab when performing this work ? Or maybe there is some reason related to the studied sample ?	example, s-SNOM resolutions down to ~5nm have been reported with inorganic specimens using sharper bespoke probes³³". We note that we are using the sharpest commercially available probes for this application.
11) Line 113: " The cell appears to be binucleated, with two protein-dense nucleoli and nuclear membranes visible in figure 4 (a)". Are myeloma cells usually binucleated ? Or this is something that is seldom observed ?	We thank the reviewer for encouraging us to provide more information here. Indeed, binucleated myeloma cells are frequently observed. We have revised the manuscript (line 133) to make comment on this and provide references: "This cell appears to be binucleated, as is frequently observed in multiple myeloma cells³⁶⁻³⁹, with two protein-dense nucleoli and nuclear membranes visible in figure 4 (a)".
12) Line 126: "By inspecting the morphology of chemical maps of cells, and comparing with supporting TEM images, we have identified many intracellular structures that have not been directly optically imaged before." With respect to the last part, it would be useful to add a table or paragraph, and summarize what cell parts discussed in this manuscript have been resolved with other optical techniques, and which haven't.	We accept the reviewer's request to be more specific here. The manuscript has been revised (line 151): "To the authors' knowledge, this paper provides the first directly optical, sub-diffraction images of endoplasmic reticulum, mitochondria, and dense fibrillar components and fibrillar centres in nucleoli".
13) Line 140: "We note that s-SNOM imaging should be suitable for samples that have been prepared for correlative light and electron microscopy³⁶, in particular resin-embedding and cryosectioning approaches." s-SNOM imaging has actually been already demonstrated on samples prepared for correlative light and electron microscopy. In [Stanciu S.G., Biomed. Opt. Express 8, 5374-5383 (2017)] the authors discussed correlative s-SNOM -CLSM imaging on fluorescently labeled ultra-thin zebrafish retina sections cut with an ultramicrotome, according to correlative fluorescence-EM protocols [J. M. Mateos, Sci. Rep. 6, 34062 (2016)].	We thank the reviewer for making this point. We have revised the manuscript to include both of these references and account for this (line 165): "Correlative s-SNOM and light microscopy¹⁵ has already been demonstrated and we expect that correlating additionally with electron microscopy is also possible with adaptations of current techniques^{44,45}".
14) Line 146: "Since the maximum achievable spatial resolution in s-SNOM is related to the size of the tip of the scanning probe¹⁷, better resolution can be expected if a sharper tip is used, albeit to the detriment of image signal-to-noise ratio. Developments leading to improved sensitivity in s-SNOM imaging, for example using 10 infrared nanoantennas⁸, might help to alleviate this signal-to-noise issue". Recently a study introduced scanning probe tips with tunable optical properties [Belhassen, J., et al.	We thank the reviewer for pointing us in the direction of this newly published work. This indeed might be applicable to our work going forward. We have included this reference in the revised manuscript (line 174).

Advanced Photonics Nexus 2.2 (2023): 026002.]. Would such tips be of benefit with respect to higher SNR in this bioimaging application here discussed ?	
15) I think not all figures in the supplementary are referenced in the text. The authors should check.	We can confirm that all figures are referenced in the text.
16) It would be relevant if the authors could comment on how qualitative interpretation of s-SNOM phase images collected on myeloma cells could potentially be complemented by quantitative s-SNOM (refractive index) imaging, which has been previously demonstrated on red blood cells. [Tranca, D.E. Nanomedicine 14.1 (2018): 47-50].	We thank the reviewer for this suggestion and agree it is relevant to include. We have included it in the revised manuscript (line 160): "Further, conversion from s-SNOM signals to commonly used optical parameters such as the refractive index is also possible through image processing algorithms ¹³ ".
17) The authors do not discuss at all amplitude s-SNOM images, nor AFM images. Are s-SNOM amplitude images useless in this application ? At least in the supplementary a full set (s-SNOM amplitude, s-SNOM phase, AFM) should be displayed. Also, it would be interesting to discuss how s-SNOM amplitude images depend on illumination wavelength. Can this dependence be physically modelled ? In the paragraph that the authors will add to the paper according to point 9 earlier, they may also refer to dependence on s-SNOM amplitude images with wavelength.	A full set of amplitude, phase and AFM images are now provided in Supplementary Figure 2. The caption details the information that can be obtained from each image.
18) In [Kanevche, K., Commun Biol 4, 1341 (2021)] the authors showed 3D s-SNOM imaging of prokaryotic cells, based on imaging of consecutive sections. This would be very nice to add to this work.	We thank the reviewer for the suggestion. Indeed, this would be a nice addition, but unfortunately we don't have the sequential sections required to implement this at this stage. We will look to include this in future projects.
19) It would be very useful for the s-SNOM community if the authors upload the whole raw dataset collected in this experiment as a public resource (e.g. on Data Dryad, OSF, FigShare, etc) and add it as a reference.	We accept the reviewer's invitation to share images publicly. They are now available at the BioImage Archive repository under accession code S-BIAD612. This is detailed in the Data Availability section of the paper.
Reviewer #3 (Remarks to the Author):	
In this manuscript, the authors apply the infrared nanoscopy technique of s-SNOM to measure tissues. IR s-SNOM obtained chemical contrast that regular electron microscopy cannot provide. The author demonstrated IR imaging of tissue based on several IR absorption bands of biological molecules. The results have their merits in the biological application of s-SNOM, which is a tool that has just started to gain traction in the realm of biology. The work is publishable.	We thank the reviewer for taking the time to review this work and for their encouraging comments.

One minor comment is that the authors used a commercial instrument of NeaSNOM, and stated that third harmonic demodulation was used. However, regular third harmonic demodulation does not give a phase value corresponding to the IR absorption. The authors must have used the technique of pseudoheterodyne, which is available in NeaSNOM. The author should update their method description to correctly state this technical detail. For those who do not have NeaSNOM, this information would be critical for repeatability.

We take this comment as an invitation to add clarity. We have revised the manuscript to put more emphasis on the necessity of using pseudoheterodyne detection for our measurements (line 69): "Critical to the technique is a pseudoheterodyne detection scheme²⁵. The signal measured at the detector is analysed at harmonics of the probe oscillation frequency, Ω , which are split into sidebands separated by the phase modulation frequency, M , of the vibrating reference mirror. For the images presented in this paper, the third harmonic of the probe oscillation frequency is used. This yields a highly surface-sensitive, background-free phase measurement, ϕ_3 ". We have also included this detail in the methods section (line 208): "A pseudoheterodyne detection scheme²⁵ was employed to obtain background-free phase measurements".

Updated / Additional figures in the supplementary material (No changes to figures in the manuscript)

1. Supplementary figure 1: To satisfy point 6, reviewer 2

New figure showing s-SNOM images at each available harmonic. See supplementary for full caption.

2. Supplementary Figure 2: To satisfy point 7, reviewer 2

New figure showing s-SNOM phase, amplitude and AFM images. See supplementary for full caption.

3. Supplementary Figure 4: To satisfy point 3, reviewer 1

Added FTIR spectrum of resin (panel a). See supplementary for full caption.

Reviewers' comments:

Reviewer #1 (Remarks to the Author):

I would like to express my appreciation for the authors' efforts to address my previous comments. While most of my concerns have been resolved, I am still unclear as to why the authors state that spectroscopy cannot be achieved without the broadband light source and nanoIR module. I noticed in Line 37 of the updated manuscript that the authors mentioned using a tunable QCL, and in Line 40, they stated that "Tuning the laser allows us to perform IR spectroscopy at the nanoscale". This seems to contradict the authors' statement about the inability to obtain spectra. Furthermore, the MIRcat QCL itself can sweep wavenumbers, which suggests that point spectroscopy is feasible.

As I was reading the revised manuscript, I also had a question regarding the background signal resulting from the resin. Figure 3b (iv) shows that the nucleolus at 1020 cm⁻¹ has a weaker signal than the resin region, and Supplementary Figure 4 confirms that resin indeed has a non-negligible absorption at this frequency. This resin background can create false positive signals when detecting weak signals, such as those from nucleic acids. Therefore, I suggest that point spectra are necessary to accurately identify the components present at each location.

Finally, I noticed a discrepancy in Supplementary Figure 2a, which is the only topography image shown in the manuscript. The estimated thickness of the cell slice in Supp. Figure 2a is around 20 nm, whereas the authors state that the thickness should be between 70 – 200 nm. I am curious about the reason for this inconsistency.

Reviewer #2 (Remarks to the Author):

The authors have provided insightful answers to my questions, and considerably enhanced the manuscript during this revision round. I am pleased to recommend for this article to be accepted in its current version.

COMMSBIO-23-0150A Reviewer comments response (2nd revision)

Reviewer comments	Response to reviewer
Reviewer #1 (Remarks to the Author):	
I would like to express my appreciation for the authors' efforts to address my previous comments.	We thank the reviewer for their further feedback on the manuscript. We have revised the manuscript and included an extra supplementary figure in response to the comments. Please see below for details.
While most of my concerns have been resolved, I am still unclear as to why the authors state that spectroscopy cannot be achieved without the broadband light source and nanoIR module. I noticed in Line 37 of the updated manuscript that the authors mentioned using a tunable QCL, and in Line 40, they stated that "Tuning the laser allows us to perform IR spectroscopy at the nanoscale". This seems to contradict the authors' statement about the inability to obtain spectra. Furthermore, the MIRcat QCL itself can sweep wavenumbers, which suggests that point spectroscopy is feasible.	Indeed the MIRcat can sweep through wavenumbers, but generating point-spectra would require us to manually retune, optimise and collect an image at each wavenumber which would take considerable time. Further, the spectrum would have to be normalised to the embedding resin which is not suitable for producing an absorption spectrum due to its non-uniform absorption. For instance, the resulting absorption spectrum would be negative at wavelengths where the resin absorbs strongly. We accept that our statement "Tuning the laser allows us to perform IR spectroscopy at the nanoscale" might lead readers to think that we can obtain absorption spectra akin to those obtained with (nano) FTIR, so we have replaced this with "Tuning the laser allows us to image with chemical specificity" (line 40).
As I was reading the revised manuscript, I also had a question regarding the background signal resulting from the resin. Figure 3b (iv) shows that the nucleolus at 1020 cm ⁻¹ has a weaker signal than the resin region, and Supplementary Figure 4 confirms that resin indeed has a non-negligible absorption at this frequency. This resin background can create false positive signals when detecting weak signals, such as those from nucleic acids. Therefore, I suggest that point spectra are necessary to accurately identify the components present at each location.	The resin indeed contributes to the s-SNOM signal at 1020cm ⁻¹ , but not strongly enough to impair our ability to image the cells with chemical specificity. We show this in a new supplementary figure (at the end of this document), in which the structures seen at 1735cm ⁻¹ and 1155cm ⁻¹ (resin absorption) are not seen at 1020cm ⁻¹ and 1240cm ⁻¹ (i.e. the distribution of resin is not introducing any artefacts in the images). Also, the strongly absorbing structures visible at 1020cm ⁻¹ are not identifiable at 1735cm ⁻¹ or 1155cm ⁻¹ , which suggests they are chemically specific and not related to the resin.
Finally, I noticed a discrepancy in Supplementary Figure 2a, which is the only topography image shown in the manuscript. The estimated thickness of the cell slice in Supp. Figure 2a is around 20 nm, whereas the authors state that the thickness should be between 70 – 200 nm. I am curious about the reason for this inconsistency.	Supplementary figure 2a doesn't show the edge of the section so the section thickness cannot be calculated. Instead it shows the topography of the section and that the thickness of the section varies by around 20nm. I have rephrased the caption to avoid any confusion, replacing "The AFM image (a) maps out the surface height of the section" with "The AFM image (a) shows the topography of the section".

Reviewer #2 (Remarks to the Author):	
The authors have provided insightful answers to my questions, and considerably enhanced the manuscript during this revision round. I am pleased to recommend for this article to be accepted in its current version.	We thank the reviewer for their encouraging feedback and agree that the manuscript has been strengthened after responding to their comments.

Additional supplementary figure (Supplementary Figure 8)

This new figure shows images of a myeloma cell nucleoli at wavelengths which map nucleic acids (a,b) and the embedding resin (c,d). Features in c and d (yellow arrows) do not appear as artefacts in a and b. Features in a and b (cyan arrows) are not identifiable in c and d, suggesting they are chemically specific to a and b.

Supplementary Fig 8. Chemical mapping of a myeloma cell nucleolus at imaging wavelengths exciting phosphodiester moieties (a), C–O bonds (b), C=O bonds in the embedding resin and fixatives (c), and C–O–C bonds in the embedding resin (d). Whilst it is likely that the resin makes a small contribution to the s-SNOM signal at the imaging wavelengths in (a) and (b), this contribution doesn't appear to introduce artefacts as features in (c) and (d) indicated with yellow arrows are not present in (a) and (b). The structures indicated by the cyan arrows in (a) and (b) appear to be chemically specific as they are not identifiable in (c) and (d). Scale bars 250nm.

REVIEWERS' COMMENTS:

Reviewer #1 (Remarks to the Author):

The authors have taken into account all of my comments and concerns, and I am pleased to recommend accepting this article in its current form.